# Recent Advances in Monoclonal Antibody Therapy for Colorectal Cancers

**DOI:** 10.3390/biomedicines9010039

**Published:** 2021-01-05

**Authors:** Kyusang Hwang, Jin Hwan Yoon, Ji Hyun Lee, Sukmook Lee

**Affiliations:** Biopharmaceutical Chemistry Major, School of Applied Chemistry, Kookmin University, Seoul 02707, Korea; kyusang@kookmin.ac.kr (K.H.); yoonjinhwan8090@kookmin.ac.kr (J.H.Y.); 707jh@kookmin.ac.kr (J.H.L.)

**Keywords:** colorectal cancer, monoclonal antibody, therapeutic target, therapy

## Abstract

Colorectal cancer (CRC) is one of the leading causes of cancer deaths worldwide. Recent advances in recombinant DNA technology have led to the development of numerous therapeutic antibodies as major sources of blockbuster drugs for CRC therapy. Simultaneously, increasing numbers of therapeutic targets in CRC have been identified. In this review, we first highlight the physiological and pathophysiological roles and signaling mechanisms of currently known and emerging therapeutic targets, including growth factors and their receptors as well as immune checkpoint proteins, in CRC. Additionally, we discuss the current status of monoclonal antibodies in clinical development and approved by US Food and Drug Administration for CRC therapy.

## 1. Introduction

Monoclonal antibody therapy is an effective therapeutic intervention to treat patients with chronic (e.g., cancers and immunological disorders) as well as acute infectious diseases [1]. Since the discovery of hybridoma technology by Kohler and Milstein in 1975 [2], OKT3, the first US Food and Drug Administration (FDA)-approved mouse monoclonal antibody specific to CD3, was developed to prevent or reverse graft rejection by blocking T-cell activation [3]. However, it is not widely used in clinics because of the immunogenicity issue observed in OKT3 treatment [4]. Consequently, the remarkable development of recombinant DNA technology created many cutting-edge technologies for the development of therapeutic antibodies, including antibody library construction, phage display, high-throughput-based antibody selection, affinity maturation, humanization, and overproduction [5,6,7,8,9,10]. For example, phage display is the most common and practical technology for peptide or antibody selection that was initially developed by. Smith and Winter, the 2018 Nobel laureates in chemistry. Antibodies, such as antigen-binding fragments (Fab) or single-chain variable fragment (scFv), are displayed on a phage that confers antibodies with the key properties of replicability and mutability [11]. Furthermore, Gregory P. Winter and his team pioneered the humanization techniques to lower the immunogenicity elicited by nonhuman monoclonal antibodies [12]. Moreover, antibody humanization is a state-of-the-art technique to humanize the variable region of the antibodies obtained from nonhuman species including mice, rabbits, and chickens. Among several approaches, the complementarity-determining region (CDR) grafting method has been mostly used in antibody humanization for the development of therapeutic antibodies including cetuximab, rituximab, and infliximab. Humanized antibodies by CDR grafting are rendered by transferring the CDRs of a variable region to a human antibody scaffold [10,13,14,15]. As of December 2019, 79 therapeutic monoclonal antibodies have been approved by the US FDA. Furthermore, the global therapeutic monoclonal antibody market is the fastest-growing pharmaceutical industrial market and is expected to generate revenue of $300 billion by 2025 [14]. In addition, several antibody fragments such as Fab and scFv have also entered clinical trials [16]. Antibody fragments retain the targeting specificity of whole monoclonal antibodies and can be more economically produced. Furthermore, these fragments are smaller and easily penetrate tissues and tumors more rapidly and deeply than monoclonal antibodies having a higher molecular weight of 150 kDa. Antibody fragments have been also forged into multivalent and multispecific reagents, linked to therapeutic payloads (e.g., radionuclides, toxins, enzymes, liposomes, and viruses), and engineered for enhanced therapeutic efficacy [16,17,18,19].

Colorectal cancer (CRC) is the third most commonly occurring malignancy and second leading cause of cancer death worldwide. The increasing prevalence of CRC across the globe is one of the key factors driving the growth of the market [20]. According to statistics, the global CRC therapeutic market is expected to reach $18.5 billion by 2023 from $13.7 billion in 2018, at a compound annual growth rate of 6.1% from 2018 to 2023 [21]. Traditionally, 5-fluorouracil (5-FU)-based chemotherapeutic regimens, such as FOLFIRI (irinotecan-containing regimen) and FOLFOX (oxaliplatin-containing regimen), have been used clinically as standard therapies for treating CRC patients [22]. Recently, monoclonal antibodies have been used in combination with standard chemotherapy to improve the clinical outcomes of CRC patients. Compared with traditionally used chemotherapeutic agents, monoclonal antibodies have fewer side-effects because their target specificity and versatility are also being applied to next-generation antibody-based therapeutics, including bispecific/multispecific antibodies, antibody-drug conjugates, and chimeric antigen receptor T cells or natural killer cells [23,24,25]. 

Over several decades, extensive in vitro and in vivo biochemical and molecular biology studies have suggested many key signaling molecules closely associated with CRC progression and metastasis. Among them, several growth factors, including epidermal growth factor (EGF), vascular endothelial growth factor (VEGF), and hepatocyte growth factor (HGF) and their cognate receptors, are proven therapeutic targets in CRCs for monoclonal antibody therapy [26,27]. Further, human EGF receptor type 2 (HER2) is also known as a monoclonal antibody target in CRCs [28]. More recently, Alison and Honjo, the 2018 Nobel prize winners in physiology, discovered immune checkpoint proteins, such as programmed cell death protein-1 (PD-1) and cytotoxic T lymphocyte antigen 4 (CTLA-4), which are key negative regulators of the immune system and cancer growth [27]. Presently, these immune checkpoint proteins are drawing attention as the most promising therapeutic targets in other types of cancers as well as CRCs for monoclonal antibody therapy. In addition, with infrastructural and technical advancement in monoclonal antibody development, blockbuster humanized and fully human monoclonal antibodies, including cetuximab, bevacizumab, and pembrolizumab, have received FDA approval and are widely used to treat CRC patients [29].

In this review, we first highlight recent studies of the roles and relevance of therapeutic targets in CRCs for monoclonal antibody therapy to understand the pathological mechanism of CRCs governed by the target molecules. Simultaneously, presenting the current status of FDA approved monoclonal antibodies in clinical development for CRC therapy will provide insight into unmet medical needs in CRCs for monoclonal antibody-based therapy.

## 2. Physiological and Pathophysiological Roles of Therapeutic Targets in CRCs

### 2.1. EGF/EGFR

EGF is a 6 kDa growth factor with 53 amino acid residues; it binds to epidermal growth factor receptor (EGFR) that is a single transmembrane glycoprotein with 1186 amino acids. ErbB family members comprise ErbB1 (EGFR, HER1), ErbB2 (HER2), ErbB3 (HER3), and ErbB4 (HER4). EGF binding to EGFR occurs within the 622-amino acid extracellular domain (ECD), which is divided into four distinct domains: I–IV. Especially, domains I and III are responsible for ligand binding, whereas domains II and IV have two cysteine-rich regions that form disulfide bonds. Further, EGFR also has a 23-amino acid residue α-helical transmembrane domain, 250-amino acid tyrosine kinase domain, and 229-amino acid C-terminal tail with regulatory tyrosine residues [30,31,32].

Under physiological conditions, EGFR activation proceeds sequentially by two steps. First, prior to ligand binding, domain II is folded into domain IV via disulfide bonds in a tethered conformation. Second, once EGF binds to domains I and III of EGFR monomers, EGFRs promote domain rearrangement to expose dimerization arms in domain II, which leads to receptor dimerization via domain II [31,33]. In turn, the EGFR dimers induce trans-autophosphorylation by tyrosine kinase domains within the cytosolic parts of each EGFR, resulting in activation of their downstream signaling cascades, such as the rat sarcoma (RAS)/rapidly accelerated fibrosarcoma (RAF)/mitogen-activated protein kinase (MAPK) and phosphoinositide 3-kinase (PI3K)/Akt pathways [33,34,35]. The RAS–RAF–MAPK pathway is a major downstream signaling route of the ErbB family. EGF binding to EGFR and consecutive tyrosine phosphorylation in EGFRs leads to activation of RAS, a small GTP-binding protein, with the help of growth factor-bound protein 2 (GRB2), an adaptor protein, and son of sevenless (SOS), a guanine nucleotide exchange factor. In turn, activated RAS then activates downstream signaling molecules, including RAF and MAPK [36]. Activated MAPKs phosphorylate specific transcription factors and participate in regulation of cell migration and proliferation. EGFR activation also stimulates PI3K composed of separate regulatory (p85) and catalytic (p110) subunits. The p85 regulatory subunit directly binds to EGFR through the interaction of its Src homology domain 2 (SH2) with phosphotyrosine residues in activated EGFR. At the same time, the p110 catalytic subunit catalyzes phosphorylation of phosphatidylinositol 4,5-diphosphate to generate phosphatidylinositol 3,4,5-triphosphate (PIP_3_), which in turn activates the protein serine/threonine kinase, Akt [37,38,39,40] (Figure 1).

Many previous reports have suggested the importance of EGFR as a therapeutic target in CRC. For example, immunohistochemical studies have shown that EGFR was highly overexpressed in 118 (80%) of 150 CRC patients, with a median follow-up of 40 months. In addition, *Balb/c* athymic nude mice subcutaneously injected with HCT116 or EGFR-knockout CT116 cells also revealed that depletion of EGFR in HCT116 cells was associated with reduced tumor growth [41]. Osimertinib (Tagrisso^®^, AstraZeneca, Cambridge, UK), a tyrosine kinase inhibitor of EGFR, was found to remarkably decrease the tumor size and growth rate in a DLD-1, a colorectal adenocarcinoma cell line, xenograft mouse model [42]. Administration of cetuximab, a human/mouse chimeric antibody, exhibited significant growth suppression and inhibited the EGFR/MAPK pathway in a HT29 xenograft mouse model [43]. Furthermore, another anti-EGFR antibody also showed similar growth inhibition of CRCs to cetuximab in preclinical settings. GC1118, a novel fully human anti-EGFR IgG1 antibody, exhibited potent inhibitory effects on EGFR signaling, enhanced antibody-mediated cytotoxicity, and significantly inhibited tumor growth in a CRC patient-derived xenograft (PDX) model [44]. Further, Ame55, an anti-EGFR IgG1 antibody, also inhibited tumor growth in a LoVo xenograft mouse model [45].

In addition, some reports show the interrelationship of EGFR with other biomarkers. Consequently, HER2 amplification has been implicated in therapeutic resistance to anti-EGFR antibody therapy in preclinical studies in metastatic CRC (mCRC) [46]. HER2 amplification is seen in a small subset of mCRC, predominantly in KRAS wild-type tumors, for which anti-EGFR antibodies are used as targeted therapies [47,48]. In these tumors, aberrant HER2 signaling results in the bypass of the activation of the RAS/MEK/MAPK signaling pathway, thereby blunting the effect of EGFR blockade [49,50]. Furthermore, the EGFR-MET interaction induced by transforming growth factor-α, a specific EGFR ligand, overexpression and concomitant phosphorylation of MET, and activation of MET downstream effectors have been proposed to be closely associated with the acquired resistance to cetuximab in CRC cells [51,52]. Thus, these pieces of evidence demonstrate that the combined inhibition of EGFR and other biomarkers can represent an effective strategy for overcoming cetuximab resistance in patients with CRCs [53].

### 2.2. VEGF/VEGFR

VEGF is a disulfide-bonded dimeric glycoprotein of 45 kDa. The VEGF family has five isotypes, including VEGF-A, placental growth factor, VEGF-B, VEGF-C, and VEGF-D [54]. Among them, VEGF-A is a glycosylated mitogen protein that is closely associated with regulation of numerous pro-angiogenic functions, including endothelial cell growth and migration, and vascular permeability in angiogenesis [55]. However, less is known about the function and regulation of VEGF-B, -C, and -D. VEGFR receptor (VEGFR) is a member of the receptor tyrosine kinase (RTK) family having multiple immunoglobulin-like ECDs and tyrosine kinase activity. VEGFRs are divided into three types: VEGFR1 (Flt-1), VEGFR2 (KDR or Flk-1), and VEGFR3 (Flt-4). Especially, VEGFR2, a 200–230 kDa protein, mostly interacts with VEGF-A and has a key role in angiogenesis at early embryogenesis and mostly at lymph angiogenesis [54,56].

Similar to that of other RTKs, the binding of VEGF to VEGFR forms a receptor dimer and induces trans-autophosphorylation. Specifically, signaling of VEGFRs is initiated upon binding of a covalently linked ligand dimer to the ECD of the receptor. This signaling promotes receptor homodimerization followed by phosphorylation of specific tyrosine residues located in the intracellular juxtamembrane domain, the kinase insert domain, and the carboxy terminal tail of the receptor. Subsequently, a variety of signaling molecules are recruited to VEGFR dimers [57,58]. These interactions next activate phospholipase C (PLC)γ and protein kinase C (PKC) to induce transcription of genes necessary for angiogenesis and cell proliferation [59]. Simultaneously, VEGFR-induced activation of PI3K results in accumulation of PIP3, which induces phosphorylation of Akt to increase endothelial cell survival and also induces systemic destruction of the entire basement membrane to increase vascular permeability [60,61]. Ligand binding to VEGFR-2 also triggers the activation of the RAS pathway, initiating signaling through the RAF–MEK–MAPK pathway known to be important in VEGF-induced cell proliferation [62]. In addition, VEGF stimulates p38 MAPK to regulate the rearrangements of the actin cytoskeleton in cell migration [63] (Figure 1). 

It has been reported that 70% of patients with stage IV CRCs had positive VEGF expression, whereas 50% and 47% of patients with stage II and III CRCs, respectively, had positive VEGF expression. A statistically significant correlation was found between VEGF and 10-year disease-specific survival: VEGF-expressing tumors were more frequent in patients who died of the disease than in those who survived for 10 years [64]. Further, other reports have shown that the VEGF level also increased in CRC patients. In a prospective study by Anastasios et al. that included 67 consecutive colorectal patients, VEGF was detectable in all control subjects. Their median serum VEGF level was 186 pg/mL. Additionally, serum VEGF levels were higher in 67 patients with newly diagnosed and histologically confirmed primary CRC (492 pg/mL) than in the control subjects (186 pg/mL) [65].

An increasing number of reports have suggested VEGF/VEGFR signaling as a promising therapeutic target in CRCs. First, Foersch et al. generated a conditional knockout for VEGFR2 to investigate the functional role and underlying molecular mechanisms of the signaling. Specific deletion of VEGFR2 was confirmed by qPCR of cDNA. Immunofluorescence staining revealed a lack of receptor expression relative to that of VEGFR2-expressing control mice. Consequently, significantly fewer tumors developed in VEGFR2-knockout mice than in control mice [66]. Second, regorafenib, a novel small-molecule multi-kinase inhibitor, markedly slowed tumor growth in five of seven PDX models. The antitumor effects of regorafenib were evaluated in seven PD CRC xenografts [67]. Third, bevacizumab, a humanized antibody to VEGF, showed a significant delay in CRC tumor growth relative to that of the non-treated animals [68]. Fourth, intraperitoneal injection of DC101, an anti-VEGFR mouse monoclonal antibody, inhibited tumor growth and induced apoptosis in CRC in a KM12L4 xenograft model. Further, treatment with DC101 decreased tumor vascularity, growth, proliferation, and increased apoptosis [69]. Taken together, this large number of studies shows the importance of VEGF/VEGFR signaling for developing pharmaceutical anti-angiogenic drugs.

### 2.3. HGF/c-MET

HGF is synthesized as an inactive precursor (pro-HGF) that undergoes site-specific proteolytic cleavage by extracellular serine proteinases into an active 90-kDa heterodimer containing α and β chains. HGF is predominantly secreted by stromal cells and activates mesenchymal-epithelial transition factor (c-MET) on adjacent epithelial cells [70]. HGF contains two c-MET binding sites; a high affinity site in the N-terminal and first kringle regions that binds to the immunoglobulin-like fold shared by plexins and transcriptional factors (IPT) 3 and IPT4 domains in c-MET and a low-affinity site in the serine protease homology domain that interacts with the semaphorin domain in c-MET [71]. As a tyrosine-protein kinase Met or HGR receptor, c-MET is synthesized as a 170-kDa single-chain precursor protein (pro-c-MET) that undergoes furin-mediated post-translational cleavage, yielding a disulfide-linked heterodimer composed of an extracellular α-subunit and a single transmembrane β-subunit [70,72].

In the physiological state, HGF binding to c-MET induces c-MET dimerization and trans-autophosphorylation of the tyrosine residues Y1003 in the juxtamembrane domain and Y1234 and Y1235 within the kinase activation loop, resulting in phosphorylation of two tyrosine residues, Y1349 and Y1356, in the C-terminus. Tyrosine phosphorylation of Y1349 and Y1356 residues creates docking sites for recruitment of key intracellular adaptor proteins and signaling molecules through SH2-mediated interactions, including GRB2, GRB2-associated binding protein 1 (GAB1), signal transducer and activator of transcription 3 (STAT3), the p85 subunit of PI3K, SRC, PLC-γ, and Shc [73,74]. The potency, duration, and versatility of HGF/c-MET signaling are modulated by signaling amplifiers and co-receptors. Recruitment and sustained phosphorylation of the multi-adaptor protein GAB1 is an important hallmark of sustained c-MET signaling that can either bind directly to c-MET through its unique 13-amino-acid c-MET binding site or indirectly by association with GRB2. The tyrosine-phosphorylated GAB1 protein serves as an auxiliary signal transduction platform through recruitment of various effector proteins, including PI3K. Cell survival response is related to the PI3K/Akt pathway activated by c-MET signaling [75,76]. Furthermore, HGF-induced c-MET-dependent RAS/RAF/MAPK activation has been found to require the co-receptor CD44v6, thus providing a platform for SOS recruitment to the complex and subsequently triggering proficient activation of RAS [32]. Invasion, branching morphogenesis, and tumorigenesis are mediated by STAT3 activation in a tissue-dependent manner. Typically, STAT3 activation is induced by phosphorylation on a critical tyrosine residue (Y705) that triggers STAT3 dimerization owing to reciprocal phosphotyrosine-SH2 domain interactions. In addition to tyrosine 705 phosphorylation, STAT3 is also activated through serine (S727) phosphorylation. Finally, the reversible acetylation of STAT3 by histone acetyltransferase on a single lysine residue (K685) represents a third mechanism of STAT3 activation. Acetylated STAT3 enhances the stability of STAT3 dimers, which are required for DNA-binding and transcriptional activity. Cellular migration and adhesion are mediated by a c-MET–SRC–focal adhesion kinase (FAK) interaction. Within the complex, Src phosphorylates Y576 and Y577 within the kinase domain activation loop and Y861 and Y925 within the C-terminal domain of FAK. The FAK–Src complex further binds to and phosphorylates various adaptor proteins, such as p130Cas and paxillin [77,78,79] (Figure 2). 

c-MET is overexpressed in CRCs [80]. Specifically, immunohistochemistry with 23 cases of colorectal adenoma and 102 cases of primary colorectal carcinoma as well their corresponding metastases (44 lymph nodes, 21 peritoneal deposits, and 16 liver metastases) showed that normal tissues had a negative or weak c-MET expression, whereas c-MET was highly overexpressed in adenomas and primary CRC. Moreover, c-MET expression in metastatic tissues was significantly higher compared with the primary tumor [81].

Currently, HGF/c-MET signaling is one of the key therapeutic targets in CRC therapy. Small hairpin RNA-mediated c-MET knockdown dramatically suppressed tumor growth in a SW480 xenograft mouse model as well as SW480 cell proliferation in vitro [82]. SU11274, an ATP-competitive inhibitor of c-MET, significantly inhibited tumor growth in a LoVo xenograft mouse model. ARQ 197 (tivantinib), a non-ATP-competitive inhibitor of c-MET, decreased tumor growth in a HT29 xenograft mouse model [83]. Antibody-based targeting of c-MET also gave results similar to those of pre-existing chemical inhibitors. For example, YYB-101, a humanized neutralizing antibody specifically binding to HGF, inhibits c-MET activation and cell scattering in vitro and suppresses tumor growth in HCT116 xenograft mouse models [84]. R13 and R28, two fully human antibodies against c-MET, synergistically inhibit HGF binding to c-MET and elicit antibody-dependent cellular cytotoxicity. The combination of R13/28 significantly inhibited tumor growth in xenograft models of various colon tumors, including OMP-C12, 27, and 28. Inhibition of tumor growth was associated with induction of hypoxia. Moreover, in an experimental metastasis model, R13/28 increased survival by preventing recurrence of otherwise lethal lung metastases [85].

### 2.4. HER2

HER2 is an ErbB family member with a molecular weight of 185 kDa comprising a 632-amino acid ECD, 22-amino acid α-helical transmembrane domain, and a 580-amino acid tyrosine kinase domain. Despite the many intensive studies on HER2, a ligand of HER2 has not been clearly identified yet. It is known that HER2 forms complexes with HER2 or other ErbB family members, including EGFR, ErbB3, and ErbB4, to activate downstream signaling pathways [86,87].

Similar to EGF/EGFR signaling pathways, it has been suggested that in normal cells, the HER2 complex formation, such as through homodimerization or heterodimerization, leads to continuous trans-autophosphorylation on tyrosine residues of HER2 and activates downstream signaling pathways, including the RAS/RAF/MAPK pathway, PI3K/Akt pathway, and PLC/PKC pathway. HER2 activation ultimately promotes cell growth, proliferation, and survival. The RAS/RAF/MAPK and PI3K/Akt pathways are the two most important and extensively studied downstream signaling pathways upon activation of HER2 receptors. A third key signaling in the network is the PLC-γ/PKC pathway. Binding of PLC-γ to phosphorylated HER2 stimulates PLC-γ activity and results in hydrolytic cleavage of phosphatidylinositol-4,5-bisphosphate (PIP_2_) to yield inositol 1,4,5-triphosphate (IP_3_) and 1,2-diacylglycerol. These second messengers are important for intracellular calcium release and activation of PKC. As a result of these signaling pathways, different nuclear factors are recruited and modulate the transcription of different genes involved in cell-cycle progression, proliferation, and survival [32,33,88,89] (Figure 2). 

In CRC, HER2 expression is varied because of many factors that influence the determination of HER2 expression, especially of the intracellular fraction of HER2. One report stated that HER2 overexpression was observed in 136 (11.4%) of 1195 CRC patients with moderately to poorly differentiated tubular adenocarcinomas. Further, HER2 overexpression correlated with shorter mean overall survival (OS) [90]. Other studies have also reported that membranous overexpression of HER2 occurs in only 5% of all CRC patients, whereas cytoplasmic HER2 overexpression is observed in a significant proportion (30%) of patients [91]. 

Several lines of evidence also support the idea that HER2 is a therapeutic target in CRCs. Tucatinib, a reversible inhibitor that binds to the ATP pocket of the internal domain of the HER2 receptor, prevents activation of HER2 signaling pathways [92]. Further, administration of tucatinib in a CRC PDX model significantly reduced tumor volume. Both H2Mab-19 and H2Mab-41, novel anti-HER2 IgG2 antibodies, significantly reduced tumor development in Caco-2 xenograft mouse models [93,94]. Treatment with Herceptin^®^ (Genentech, San Francisco, CA, USA) caused a decrease in HER-2 protein levels in DLD-1, HT-29, Caco-2, and HCA-7 colon cancer cells in vitro. Treatment of athymic mice engrafted with EGFR-dependent colon cancers, including HCA-7, DLD-1, and HT-29 with Herceptin^®^ showed tumor regression and decreased EGFR tyrosine phosphorylation in tumor cells [95].

### 2.5. Immune Checkpoint

#### 2.5.1. CTLA-4

CTLA-4, also designated CD152, is a type I transmembrane T-cell inhibitory molecule that functions as an immune checkpoint, downregulates immune responses and is found as a covalent homodimer of 41–43 kDa [96]. CTLA-4 is a member of the IgG superfamily that is expressed by activated T cells. CTLA-4 contains an ECD with one Ig-like V-type domain, a transmembrane domain, and cytoplasmic tail. CTLA-4 is homologous to the T-cell co-stimulatory protein, CD28, and both molecules bind to B7-1/B7-2 on antigen-presenting cells [97]. CTLA-4 binds CD80 and CD86 with greater affinity and avidity than CD28, thus enabling it to outcompete CD28 for its ligands. CTLA-4 transmits an inhibitory signal to T cells, whereas CD28 transmits a stimulatory signal [98,99].

CTLA-4 is upregulated in a manner dependent on TCR stimulation. At the cell membrane, CTLA-4 undergoes dimerization, and each CTLA-4 dimer can bind two independent B7-1/B7-2 homodimers, forming a linear zipper-like structure between B7-1/B7-2 and CTLA-4 homodimers [100]. Activated CTLA-4 binds to PI3K, the tyrosine phosphatases (SHP1 and SHP2) and the serine/threonine phosphatase PP2A. SHP1 and SHP2 dephosphorylate TCR-signaling proteins, whereas PP2A targets phosphoserine/threonine residues and is known to interfere with the activation of Akt [101].

CTLA-4 can inhibit T-cell responses by several mechanisms. One mechanism involves antagonism of B7-CD28–mediated co-stimulatory signals by CTLA-4. The fact that CTLA-4 has a much higher affinity for B7 than CD28 supports the notion that the CTLA-4–mediated sequestration of B7 is closely associated with negative regulation of T-cell signaling [102]. Another mechanism for the inhibitory activity of CTLA-4 is related to direct interaction with the TCR–CD3 complex at the immunological synapse for negative regulation of downstream signaling after TCR activation [103]. When CTLA-4 interacts with the ITAMs present on the TCR–CD3 complex, the activated CTLA-4 binds to tyrosine phosphatases, including SHP1, SHP2, and PP2A, and eventually deactivates various downstream signaling molecules of activated T cells, including zeta-chain-associated protein kinase 70, spleen tyrosine kinase, and proto-oncogene tyrosine-protein kinase [104,105,106] (Figure 3). 

Many previous reports have shown that CTLA-4 is a key therapeutic target in CRCs [107]. First, Long et al. used the CRISPR-Cas9 system to generate CTLA-4 knockout cytotoxic T lymphocytes (CTLs) and evaluated the effect on the antitumor activity of the CTLs [107]. The HCT-116 xenografted mice treated with CTLA-4 KO CTLs demonstrated repressed tumor growth and prolonged survival relative to those in the control group. All of the mice in the control group died from progressive tumors within 62 days. In contrast, only 10% of CTLA-4 KO CTLs treated mice died within that time [108]. Second, Fu et al. validated the efficacy of anti-CTLA-4 mouse monoclonal antibodies on tumor size in mice inoculated with CT26 cells. The tumor volumes were 2106 ± 205 mm^3^ on day 17 in the control group treated with vehicle only but were 23 ± 4 mm^3^ in the group treated with the anti-CTLA-4 antibodies on day 5, which indicated a statistically significant difference in antitumor activity between the treated and vehicle groups [109]. Third, Lute et al. also reported that anti-human CTLA-4 human monoclonal antibodies-treated mice survived longer than the control Ig-treated mice in a human peripheral blood leukocytes-SCID mouse model. Fourth, administration of 9H10, an anti-murine CTLA monoclonal antibody, as monotherapy moderately inhibited growth and metastatic spread of the colon cancer cells in an orthotropically implanted CT26 xenograft mouse model [110]. Further, the sole CTLA-4 inhibition significantly increased intratumoral CD8^+^ and CD4^+^ T cells and reduced FOXP3^+^/CD4^+^ Treg cells, which was associated with increased expression levels of the pro-inflammatory Th1/M1-related cytokines IFN-γ, IL-1α, IL-2, and IL-12 [111].

In summary, the evidence from the studies above shows that CTLA-4 blockade exerts inhibitory effects on growth and metastasis of CRCs.

#### 2.5.2. PD-1/PD-L1

PD-1, also designated CD279, is a 55 kDa membrane protein consisting of an ECD followed by a transmembrane region and an intracellular tail containing two phosphorylation sites located in an immunoreceptor tyrosine-based inhibitory motif (ITIM) and an immunoreceptor tyrosine-based switch motif (ITSM). Programmed cell death-ligand 1 (PD-L1, designated CD274, B7-H1) is a 44 kDa transmembrane protein expressed on T cells, B cells, macrophages, and dendritic cells as well as on tumor cells, including CRCs [112,113].

Under physiological conditions, when T cells recognize antigens on major histocompatibility complex of the target cell, inflammatory cytokines, such as tumor necrosis factor alpha and interferon gamma (IFNγ) are produced to initiate inflammatory processes. These cytokines upregulate the expression of PD-L1 on tissues and PD-1 on T cells. In turn, PD-1 directly interacts with PD-L1 to negatively regulate T-cell receptor (TCR) signaling, inhibits interleukin-2 (IL-2) production in T cells, and increases T-cell apoptosis [114,115]. Specifically, this PD-1/PD-L1 interaction induces lymphocyte-specific protein tyrosine kinase-induced phosphorylation of two tyrosine-based motifs within ITIM and ITSM of the cytoplasmic tail of PD-1. The recruitment of Src homology 2 (SH2) domain-containing protein tyrosine phosphatase 1 (SHP1) and SHP-2 phosphatase then induces dephosphorylation of the TCR signalosome, including CD3ζ, ZAP70, and PI3K kinases, resulting in the deactivation of downstream signaling targets [116]. Moreover, the PD-1/PD-L1 interaction also downregulates the protein (casein) kinase 2 expression that phosphorylates the regulatory domain of phosphatase and tensin homolog (PTEN) and inhibits phosphatase activity to remove PIP3 produced by PI3K [112,117,118]. Thus, PTEN can terminate PI3K activities by dephosphorylating PIP3, which eventually leads to immune tolerance, a phenomenon in which the immune system loses the control to mount an inflammatory response even in the presence of actionable antigens [117,119] (Figure 3). 

PD-L1 has been reported to be overexpressed in CRC. More specifically, among the 80 tumor specimens, 22 (27.5%) showed high PD-L1 expression, 24 (30.0%) showed moderate expression, and 34 (42.5%) showed weak or no PD-L1 staining. Furthermore, the high PD-L1 expression in normal tissues was observed in four (6.3%) cases [120].

Many studies have suggested PD-1 and PD-L1 as promising therapeutic targets in CRCs. First, in *BALB/c Rag2−/−γc−/−* mice engrafted with PD-L1-overexpressing and PD-L1-knockout CT26 murine colon cancer cells, Gordon et al. found that after 3 weeks, tumors were significantly smaller in the PD-L1-knockout group than in the PD-L1 overexpression group [121]. Second, Cai et al. examined the efficacy of anti-mouse PD-1 rat immunoglobulin (Ig) G2 antibodies on tumor growth in a CT26 colon cancer xenograft mouse model. The antibody treatment showed significant inhibition of transplanted-tumor growth in mice [122]. Third, in a humanized CRC PDX model established by Capasso et al., treatment with nivolumab, a fully human IgG4 (S228P) monoclonal antibody to PD-1, led to significant tumor growth inhibition coupled with increased numbers of IFNγ-producing human CD8^+^ tumor-infiltrating lymphocytes [123]. Fourth, Stewart et al. reported that MEDI4736, a human IgG1 monoclonal antibody that binds with high affinity and specificity to PD-L1, significantly inhibited the growth of human tumors in a novel CT26 xenograft model containing co-implanted human T cells. This activity is entirely dependent on the presence of transplanted T cells. Further, anti-mouse PD-L1 significantly improved survival of mice implanted with CT26 CRC cells [124]. The antitumor activity of anti-PD-L1 was enhanced by combination with oxaliplatin, which resulted in increased release of high-motility group box 1 within CT26 tumors [125].

## 3. Current Status of Monoclonal Antibodies for CRC Therapy

### 3.1. Cetuximab

Cetuximab (Erbitux^®^), developed jointly by Merck KGaA (Darmstadt, Germany) and Imclone Systems (New York City, NY, USA), is a monoclonal antibody that binds to the ECD of the EGFR. It is a human/mouse chimeric IgG1 antibody that consists of the variable fragments (Fvs) of a murine anti-EGFR antibody and human constant heavy and kappa light chains [126,127].

Cetuximab was originally known as a blockade for inhibiting interactions between EGFR and all known EGFR ligands by specifically binding to domain III of the EGFR ECD [128]. Furthermore, its binding to EGFR is also able to promote receptor internalization and concomitantly downregulate EGFR protein levels expressed on the cell surface, resulting in suppression of EGFR-dependent downstream signaling pathways and transcription. In addition to these specific modes of action, cetuximab indirectly attacks cancer cells through antibody-dependent cell-mediated cytotoxicity (ADCC). After its binding to EGFR, the IgG1 portion of cetuximab may be recognized by Fcγ receptors (FcγR) on immune effector cells, such as natural killer cells and T cells, and participates in cancer cell death. In general, FcγRs bind effectively to IgG1 and IgG3 antibodies. Thus, cetuximab is speculated to more likely stimulate ADCC than panitumumab having IgG2. Consequently, cetuximab reduces tumor angiogenesis, invasiveness, and metastatic spread [129,130,131,132].

In 2004, cetuximab received FDA approval for metastatic CRCs and head and neck cancers, and its use is recommended in combination with standard chemotherapy for treatment of patients with metastatic CRCs having EGFR-positive and wild-type KRAS (Table 1). According to the CRYSTAL clinical trial (NCT00154102), compared with FOLFIRI alone, cetuximab plus FOLFIRI improved progression-free survival (PFS) from 8.0 to 9.0 months and OS from 20 to 23.5 months for treatment of patients with KRAS wild-type [133].

### 3.2. Panitumumab

Panitumumab (Vectibix™) originally developed by Abgenix Inc. (Freemont, CA, USA) is a fully human IgG2 monoclonal antibody that specifically binds to the ECD of EGFR. Especially and different from cetuximab, it can also bind to a single-point mutation in domain III of EGFR (S468R) that confers acquired or secondary resistance only to cetuximab-treated patients [134].

In 2006, panitumumab was approved by the US FDA for treatment of patients with EGFR-expressing metastatic CRCs with disease progression or following fluoropyrimidine-, oxaliplatin-, and irinotecan-containing regimens (Table 1). Later, it was also approved for treatment of patients with refractory metastatic CRCs having EGFR-positive and wild-type KRAS. Currently, panitumumab was the first monoclonal antibody to use KRAS as a predictive biomarker [135].

Panitumumab is being used in combination with chemotherapy in the first- and second-line treatment of metastatic CRCs [136]. In the PRIME (NCT00364013) clinical trial, compared with chemotherapy alone, a first-line treatment of panitumumab plus FOLFOX improved PFS from 8.0 to 9.6 months and OS from 12 to 14 months. Further, in the 20,050,181 clinical trial (NCT00339183), compared with chemotherapy, a second-line treatment of panitumumab plus FOLFIRI improved PFS from 4 to 6 months and OS from 19 to 24 months [137,138].

### 3.3. Bevacizumab

Bevacizumab (Avastin^®^) developed by Genentech (South San Francisco, CA, USA) is a humanized IgG1 monoclonal antibody that binds to segment β5–β6 of VEGF165, known as VEGF-A; its binding to VEGF-A inhibits angiogenesis by specifically inhibiting the interaction between VEGF-A and VEGFR2. Thus, bevacizumab inhibits angiogenic signaling caused by the interaction of VEGF-A and VEGFR2 [139,140,141].

In 2004, bevacizumab received FDA approval for first- or second-line treatment with 5-FU-based therapy for patients with mCRCs (Table 1). As the first anti-angiogenic antibody drug, bevacizumab is currently being used in clinics for treatment of patients with NSCLC, mRCC, epithelial ovarian cancer, and recurrent glioblastoma as well as metastatic CRCs. In the clinical trial ECOG3200 (NCT00069095) for treatment of patients with mCRCs, compared with FOLFOX4 alone, bevacizumab plus FOLFOX4 improved OS from 10.8 to 12.9 months and PFS from 4.7 to 7.3 months [142,143,144].

### 3.4. Ramucirumab

Ramucirumab (Cyramza^®^) developed jointly by ImClone Systems (New York City, NY, USA) and Dyax (Cambridge, MA, USA) is a fully human monoclonal IgG1 antibody that binds to the ECD of human VEGFR2, which is a key receptor that mediates angiogenesis and is highly expressed in not only tumor microvessels but also malignant tumors. Bevacizumab strongly neutralizes VEGF and blocks binding to VEGFR1/VEGFR2, whereas ramucirumab specifically blocks the VEGF/VEGFR2 interaction by binding to VEGFR2 [145,146,147].

Ramucirumab was isolated from a Dyax’s phage antibody library and developed as a therapeutic antibody for treatment of solid tumors. In 2014, ramucirumab first received FDA approval as a single-agent treatment for advanced gastric or gastro-esophageal junction adenocarcinoma after prior treatment with fluoropyrimidine- or platinum-containing chemotherapy. In 2015, ramucirumab in combination with FOLFIRI was also approved by the US FDA for the treatment of patients with mCRC with disease progression on or after prior therapy with bevacizumab, oxaliplatin, and fluoropyrimidine (Table 1). In the RAISE clinical trial (NCT01183780) for treatment of mCRC patients, compared with FOLFIRI alone, ramucirumab plus FOLFIRI improved OS from 11.7 to 13.3 months and PFS from 4.5 to 5.7 months [148,149].

### 3.5. Rilotumumab

Rilotumumab (AMG-102) developed by Amgen Inc. (Thousand Oaks, CA, USA) is a fully human monoclonal IgG2 antibody that binds to the beta chain of HGF. HGF, a scattering factor, influences cancer cell proliferation, survival, invasion, and metastasis through interaction with c-MET. Therefore, specific binding of rilotumumab to HGF neutralizes the interaction between HGF and c-MET and inhibits c-MET phosphorylation and downstream signaling, resulting in inhibition of cancer cell proliferation, survival, and invasion through partial antagonism of c-MET phosphorylation [150,151].

It has been reported that in the phase II clinical trial (NCT00788957) for treatment of patients with wild-type KRAS metastatic CRCs, compared with panitumumab alone, rilotumumab plus panitumumab combination therapy improved OS from 8.6 to 9.6 months and PFS from 3.7 to 5.2 months [152] (Table 1).

### 3.6. Onartuzumab

Onartuzumab (MetMAb) developed by Genentech (South San Francisco, CA, USA) is a humanized monoclonal antibody that binds to the semaphorin domain of c-MET; it is a human IgG1 with a monovalent arm. c-MET is an RTK through which downstream signals are transduced after dimerization or oligomerization by HGF binding. HGF/c-MET signaling has also been implicated in the metastatic growth of multiple cancers, making it an attractive target for various therapeutic agents [153]. When onartuzumab binds to c-MET, it inhibits cell proliferation and survival, cell motility, migration, and invasion [154]. In a clinical trial (NCT01418222) of the MET inhibitor onartuzumab in combination with mFOLFOX-6 plus bevacizumab for treatment of mCRC patients, no significant differences existed in PFS or OS between onartuzumab combination therapy with mFOLFOX-6 plus bevacizumab and bevacizumab plus mFOLPOX-6 therapy. Therefore, the phase II clinical trial of onartuzumab failed because it did not show any clinical improvement [155] (Table 1).

### 3.7. Trastuzumab

Trastuzumab (Herceptin^®^) developed by Genentech (South San Francisco, CA, USA) is a humanized IgG1 monoclonal antibody that binds to domain IV in the ECD of the HER2. This antibody is composed of humanized Fvs of murine anti-HER2 antibody and human constant heavy and kappa light chains. The antibody suppresses cancer growth and proliferation [156].

Trastzumab binds to HER2 on the cell surface of CRCs and inhibits downstream signaling pathways, such as RAS/RAF/MEK and PI3K/Akt pathways, thereby inhibiting the proliferation and survival of CRCs. This demonstrates that trastuzumab can be used as a therapeutic agent in HER2-overexpressed CRCs. In addition, trastuzumab indirectly attacks cancer cells through Fc effect functions, such as ADCC and complement-dependent cytotoxicity (CDC) [157,158].

In 1998, trastuzumab first received US FDA approval for the treatment of patients with metastatic HER2-overexpressing breast cancers, and its indications for numerous cancer treatments have expanded [159,160]. Despite the low-expression pattern observed in CRC patients, intriguingly, recent preclinical trials have shown that trastuzumab or pertuzumab in combination with lapatinib significantly suppressed tumor growth in HER2-amplified CRC tumor xenograft animal models. According to the HERACLES clinical trial (NCT03365882), compared with chemotherapy alone, trastuzumab plus lapatinib therapy in patients with HER2-positive, wild-type KRAS metastatic CRC improved PFS by 2.9 months and OS by 11.5 months [161] (Table 1).

### 3.8. Pertuzumab

Pertuzumab (Perjeta^®^) developed by Genentech (South San Francisco, CA, USA) is a humanized IgG1 monoclonal antibody that binds to HER2. The antibodies act as a blockade to inhibit dimerization of HER2 with other HER receptors, especially HER3, by specifically binding to domain II in the ECD of HER2, a different epitope for trastuzumab, resulting in inhibition of cell growth and initiation of apoptosis [162]. In 2012, the US FDA approved pertuzumab use in combination with trastuzumab and docetaxel for treatment of patients with metastatic HER2-positive metastatic breast cancers. According to the HERACLES study (NCT03365882), the randomized phase II trial studies is ongoing by evaluating the efficacy of pertuzumab and tratuzumab in patients with HER-amplified mCRCs, compared with cetuximab and irinotecan hydrochloride [163] (Table 1).

### 3.9. Ipilimumab

Ipilimumab (Yervoy^®^) developed by Medarex (Princeton, NJ, USA), is a fully humanized monoclonal antibody that binds to the ECD of CTLA-4, specifically blocks the interaction between CTLA-4 and B7-1 or B7-2, and eventually maintains T-cell cytotoxicity to attack cancer cells [164,165].

In 2011, ipilimumab first received FDA approval for melanoma treatment. Furthermore, it underwent clinical trials for the treatment of NSCLC, SCLC, and bladder cancer. Moreover, it was approved by the FDA for the treatment of mCRC in 2018 [166] (Table 1). This antibody is currently being evaluated in the CheckMate-142 clinical trial (NCT02060188) for treatment of patients with dMMR/MSI-H mCRC. The CheckMate-142 trial that is evaluating the combination therapy of ipilimumab plus nivolumab is the same clinical trial that is evaluating nivolumab [167].

### 3.10. Tremelimumab

Tremelimumab (CP-675) developed by AstraZeneca (Cambridge, UK) is a fully human monoclonal IgG2-kappa antibody that binds to the ECD of CTLA-4 and has an epitope similar to that for tremelimumab and ipilimumab. Its binding to CTLA-4 specifically blocks the interaction between CTLA-4 and B7-H1 and B7-H2 and downregulates the immune system. The function of tremelimumab is similar to that of ipilimumab, which maintains T-cell cytotoxicity against cancer cells [166,168].

In 2015, tremelimumab first received US FDA approval as an orphan drug for the treatment of patients with malignant mesothelioma. Currently, the indications that can be treated through clinical trials are expanding. In clinical trials, tremelimumab has been administered in various combination therapies with durvalumab and anti-cancer drugs. In a phase II clinical trial (NCT02870920) that compared the combination therapy of tremelizumab plus durvalumab with best supportive care for treatment of patients with advanced mCRC, the PFS was estimated to have improved to 1.8 months and the OS to have improved to 6.6 months after a median follow-up of 15.2 months (Table 1). This clinical trial is ongoing until December, 2020 [169,170].

### 3.11. Pembrolizumab

Pembrolizumab (Keytruda^®^) developed by LifeArc (London, UK) is a humanized monoclonal antibody that binds to PD-1. This antibody was generated by grafting the variable region sequences of a high affinity mouse anti-human PD-1 antibody onto a human IgG4-kappa isotype framework containing a stabilizing S228P Fc mutation for preventing Fab-arm exchange. The antibody specifically binds the PD-1 C’D-loop and antagonizes the interaction between PD-1 and its known ligands, PD-L1 and PD-L2 [171,172].

In 2014, pembrolizumab received FDA approval as the first PD-1/PD-L1 blockade drug for the treatment of metastatic melanoma [173]. In 2020, the FDA also approved pembrolizumab as the first-line treatment for patients with unresectable or metastatic microsatellite high-instability or mismatch repair-deficient CRCs [174] (Table 1). According to the KEYNOTE-177 clinical trial (NCT02563002) for treatment of MSI-H/dMMR CRCs, pembrolizumab improved PFS from 8.2 to 16.5 months compared with the standard chemotherapy [175].

### 3.12. Nivolumab

Nivolumab (Opdivo^®^) developed by Medarex (Princeton, NJ, USA) is a fully human monoclonal antibody that binds to the IgV domain of PD-1. The antibody has a variable region grafted into the human kappa and IgG4-constant region containing an S228P mutation in the hinge region. Especially, nivolumab specifically binds to the N-terminal loop of PD-1 different from an epitope for pembrolizumab. Nivolumab is also an immune checkpoint inhibitor that potentiates the cytotoxicity of T cells to kill malignant tumors [176,177].

In 2014, nivolumab first received US FDA approval for treatment of patients with advanced melanoma and was then approved to treat lung cancer in 2015. Currently, immunotherapy using nivolumab is widely used to treat mCRC [178]. In the CheckMate-142 phase II clinical trial (NCT02060188) for treatment of patients with dMMR/MSI-H mCRC, the PFS was estimated to be 12 months and the OS rate improved from 73% to 85% compared with that of nivolumab monotherapy as the primary endpoint after a median follow-up of 13.4 months (Table 1). This CheckMate-142 clinical trial is ongoing until primary completion in July 2022 [179,180].

### 3.13. Camrelizumab

Camrelizumab (AiRuiKa) developed by Jiangsu HengRui Medicine Co. Ltd. (Lianyungang, Jiangsu, China) is a humanized monoclonal antibody that binds to the flexible N- and C’D-loops of PD-1 and specifically blocks the interaction between PD-L1 and PD-1. It has epitopes partly overlapped with epitopes that bind to nivolumab and pembrolizumab. The cancer cell-killing mechanism of camrelizumab in immunotherapy is similar to that of pembrolizumab and nivolumab [181,182].

In 2019, camrelizumab first received China Food & Drug Administration approval for treatment of patients with classical Hodgkin’s lymphoma. The indications for camrelizumab are expanding through multiple ongoing clinical trials. A phase II clinical trial (NCT03912857) for treatment of patients with advanced mCRC, the combination of camrelizumab plus apatinib, a VEGFR2 inhibitor, is expected to increase the overall response rate of advanced mCRC after standard chemotherapy [183,184] (Table 1).

### 3.14. Atezolizumab

Atezolizumab (Tecentriq^®^) developed by Genentech (South San Francisco, CA, USA) is a monoclonal antibody that binds to the IgV domain of PD-L1, a ligand of the PD-1 receptor expressed on the surface of T cells. It is a humanized antibody in which the variable region of atezolizumab is changed to a human germline sequence and Fc region of atezolizumab engineered to reduce the Fc-mediated effector functions. PD-L1 acts as a T-cell suppressor through interaction with PD-1. Atezolizumab specifically binds to PD-L1 and blocks the interaction between PD-1 and PD-L1, thereby maintaining the anti-cancer effect of T cells [185,186].

In 2016, atezolizumab received FDA approval as an immune checkpoint PD-L1 inhibitor for the treatment of patients with locally advanced or metastatic urothelial carcinoma [10,16]. In 2019, the FDA also approved atezolizumab for the first-line treatment of adult patients with extensive-stage small-cell lung cancer in combination with carboplatin and etoposide, (NCT02763579) [19]. According to a clinical trial (NCT02873195) for treatment of refractory mCRC, compared with bevacizumab plus capecitabine therapy, bevacizumab and capecitabine combination therapy improved PFS from 3.3 to 4.3 months. Additionally, OS was maintained for 20 months, a result similar to that for bevacizumab and capecitabine combination therapy [155] (Table 1).

### 3.15. Avelumab

Avelumab (Bavencio^®^) developed by Merck KGaA (Darmstadt, Germany) is a fully human IgG1 antibody that binds to the IgV domain of PD-L1 in a manner similar to atezolizumab and blocks the interaction between PD-L1 and PD-1. Avelumab consists of human antibody sequences in both the variable and constant lambda light-chain and IgG1 heavy-chain regions [187].

In 2017, avelumab first received US FDA approval for the treatment of metastatic Merkel cell carcinoma [17]. Clinical trials for treatment of CRC patients ongoing. Specifically, avelumab was administered in clinical trials for CRC patients with POLE mutation observed in 3% of total CRC patients. One trial is a phase II clinical trial of avelumab monotherapy (NCT03150706) being conducted at Asan Medical Center (AMC) in South Korea. The other trial is a phase III clinical trial of avelumab plus 5-FU combination therapy (NCT03827044) being conducted at Royal Marsden Hospital in the UK that is expected to end in 2024 [188] (Table 1).

### 3.16. Durvalumab

Durvalumab (Imfinzi^®^) developed by Medimmune (Gaithersburg, MD, USA) is a fully human monoclonal antibody that binds to the ECD of PD-L1 and blocks the interaction between PD-L1 and PD-1. However, it has been reported that an epitope of PD-L1 for durvalumab is different from that of avelumab. It was engineered to prevent cytotoxic effector functions (ADCC or CDC) against PD-L1-positive immune cells [189]. Durvalumab is an immune checkpoint inhibitor used to promote immune responses for cancer therapy [189,190].

In 2017, durvalumab first received FDA approval for the treatment of metastatic urothelial carcinoma and is currently expanding indications through combination therapy [191]. The clinical trials using durvalumab are for refractory mCRC or mutated mCRC. The ILOC-EORTC phase II clinical trial (NCT030101475) for treatment of patients with refractory mCRC is ongoing with duvalumab plus tremelimumab combination therapy after radiation therapy as a primary completion in August 2021. Another phase II clinical trial (NCT03435107) for treatment of patients with POLE-mutated mCRC is ongoing to compare duvalumab monotherapy with chemotherapy. That trial has a primary completion date of March 2022 [192,193] (Table 1). 

## 4. Conclusions

CRC is a highly complex and molecularly heterogeneous disease harboring frequent mutations that are resistant to common treatment. Despite recent advances in identification of therapeutic targets in CRCs and concomitant development of their therapeutic antibodies, several medical needs remain unmet in CRC therapy. As indicated by cetuximab treatment, antibody-drug resistance is one of the major hurdles for treating CRC patients. Cetuximab is only responsive to wild-type EGFR- and KRAS-expressing CRC patients who represent 10–20% of all CRC patients, whereas cetuximab is not responsive to approximately 80–90% of CRC patients harboring gene mutations in downstream EGFR effectors, including KRAS, PI3K catalytic subunit alpha (PI3KCA), PTEN, and BRAF. Furthermore, the 5-year survival rate of approximately 13% for stage IV CRC patients highlights the importance of basic studies for understanding CRC progression and metastasis. Therefore, to improve CRC patient clinical outcomes, understanding the pathological mechanisms and identification of novel therapeutic targets in CRC remain important for developing novel monoclonal antibodies for effective CRC therapy. Although this current review focused on discussing the roles and mechanisms of monoclonal antibodies in clinical development and approved by US FDA and their cognate therapeutic targets in CRC, the therapeutic potentials of bispecific antibodies and antibody-drug conjugates have also been validated for CRC therapy based on recent increasing studies. Lastly, scientific cooperation to create scientific knowledge and successful partnership between the industry and the academia will accelerate the development of an innovative new drug for effective CRC therapy.

## Figures and Tables

**Figure 1 biomedicines-09-00039-f001:**
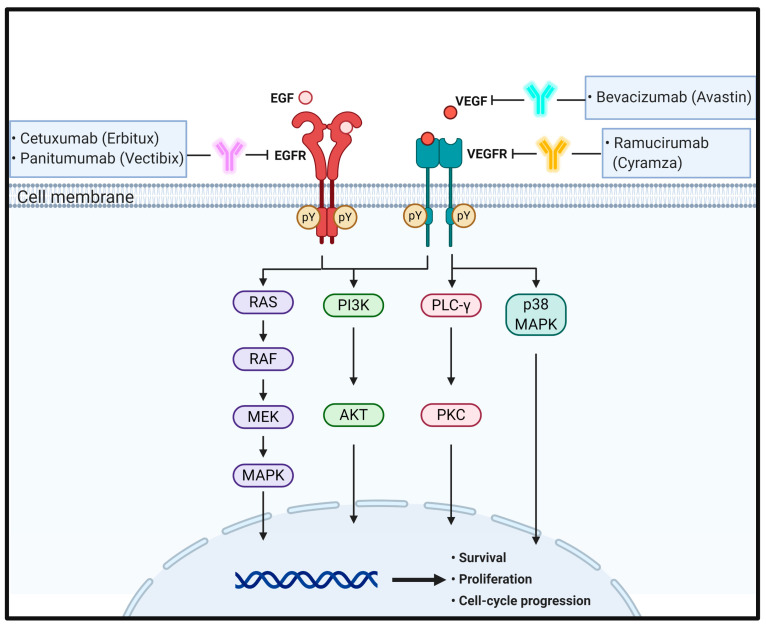
Schematic representation of the physiological roles and signaling pathways of EGF and VEGF and their cognate receptors and the effect of antibodies targeting these signaling molecules on CRC progression and metastasis. pY means phosphotyrosine residues.

**Figure 2 biomedicines-09-00039-f002:**
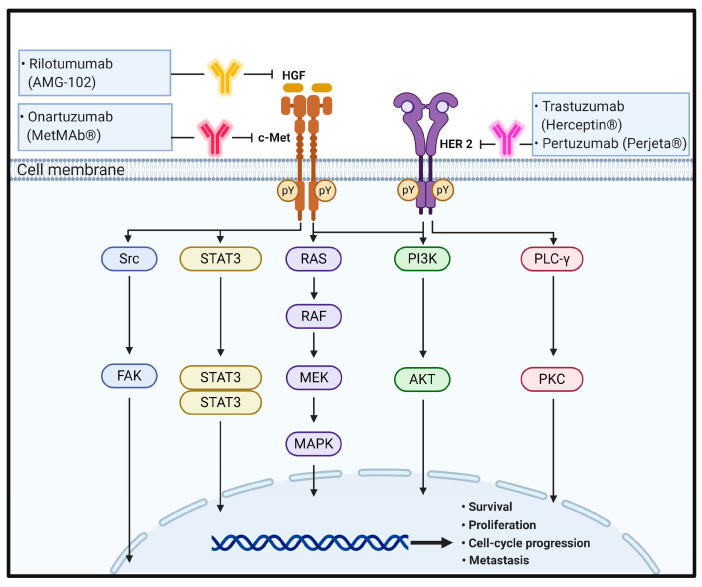
The schematic representation of the physiological roles and signaling pathways of HGF/c-MET and HER2 and effect of antibodies targeting these signaling molecules on CRC progression and metastasis. pY means phosphotyrosine residues.

**Figure 3 biomedicines-09-00039-f003:**
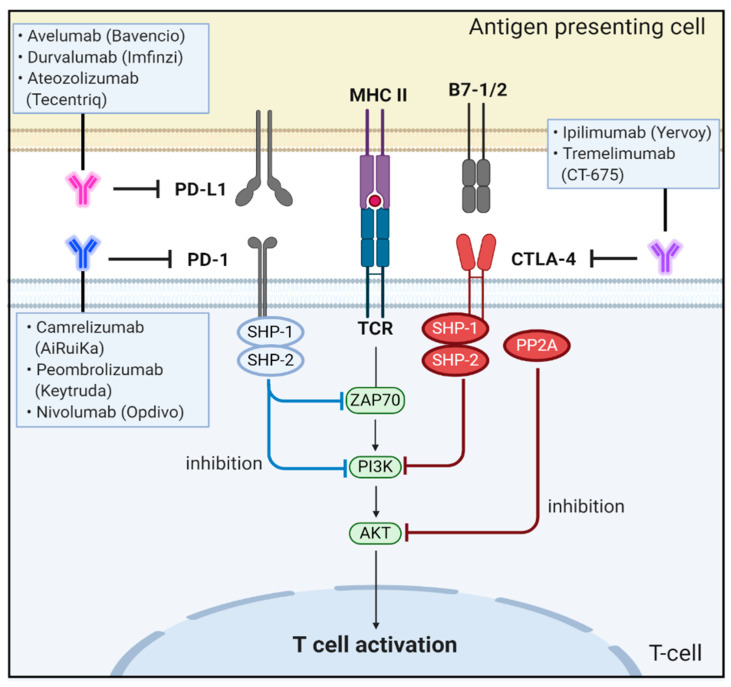
Schematic representation of the physiological roles and signaling pathways of immune checkpoint proteins of PD-1, PD-L1, CTLA-4, and B7-1/2 and effect of antibodies targeting these signaling molecules on CRC progression and metastasis.

**Table 1 biomedicines-09-00039-t001:** Antibody therapeutics currently in clinical trials or approved by the US FDA.

Name	Trade Name	Company	Target	Format	Clinical Stage
Cetuxumab	Erbitux^®^	Imclone Systems	EGFR	Chimeric IgG1	FDA approval in 2004
Panitumumab	Vectibix^®^	Abgenix Inc.	EGFR	Human IgG2	FDA approval 2006
Bevacizumab	Avastin^®^	Genentech	VEGF-A	Humanized IgG1	FDA approval in 2004
Ramucirumab	Cyramza^®^	ImClone Systems	VEGFR2	Human IgG1	FDA approval in 2015
Rilotumumab	AMG-102	Amgen Inc.	HGF	Human IgG1	P II
Onartuzumab	MetMab	Genentech	c-MET	Humanized IgG1, monovalent	P II, failure
Trastuzumab	Herceptin^®^	Genentech	HER2	Humanized IgG1	P II
Pertuzumab	Perjeta^®^	Genentech	HER2	Humanized IgG1	P II
Ipilimumab	Yervoy^®^	BMS	CTLA4	Human IgG1	FDA approval in 2018
Tremelimumab	CT-675	Medimmune	CTLA4	Human IgG2	P II
Pembrolizumab	Keytruda^®^	LifeArc	PD-1	Humanized IgG4	FDA approval in 2020
Nivolumab	Opdivo^®^	Ono Phar. & Medarex	PD-1	Humanized IgG4	P II
Camrelizumab	AiRuiKa	Jiangsu HengRui	PD-1	Humanized IgG4	P II
Atezolizumab	Tecentriq^®^	Genentech	PD-L1	Human IgG1	P II
Avelumab	Bavencio^®^	Merck KGaA	PD-L1	Human IgG1	P III
Durvalumab	Imfinzi^®^	Medimmune	PD-L1	Human IgG1	P II

Antibody information was obtained from the FDA Label database and the Drug Approval and Databases site maintained by the US FDA (https://www.fda.gov/drugs/development-approval-process-drugs/drug-approvals-and-databases) or Clinical Trial Information Site (https://clinicaltrials.gov).

## Data Availability

The data presented in this study are available on request form the corresponding author.

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
