# Peer review of "Recent Advances in Monoclonal Antibody Therapy for Colorectal Cancers"

_biomedicines, 2021, doi:10.3390/biomedicines9010039_

Round 1
Reviewer 1 Report
The review is useful and informative, but surprising scant on references. There are whole antibodies and whole biomarkers with 1-2 references when there has been a lot of work performed on the antibodies.
It would be important to consider the biomarkers relation to each other, for example, what would Her2-cMet and Ron (which is not really covered) have in overall resistance by each individual biomarker. Also what about triple negative cancers? For a start, both Trastuzumab and Pertuzumab are used together with synergy https://doi.org/10.1186/bcr2888 that was later found to be due to additional tagging https://doi.org/10.1038/npjbcancer.2015.12 that deserves discussion given the partial discussion at 10.3390/cancers10100342
There is a slight part in the beginning that discusses some antibody methods, but lacks discussion. For example can discuss further on antibody fragments e.g. https://doi.org/10.3389/fimmu.2017.01287 and humanization https://doi.org/10.1093/abt/tbaa005 , https://doi.org/10.1038/s41598-017-18892-9
It would be good to consider the biomakers and therapeutics in some chronological or order and how they can interact and work synergistically to aid in anticancer targeting.
Author Response
Response to reviewers
We appreciate the detailed and helpful comments from the reviewers. Below, we have provided point-by-point responses to thoroughly address the concerns of each reviewer. We also performed intensive proofreading by professional proofreading company. In the revised manuscript, all of modified words and sentences are in red.
▪ Comments from Reviewer #1
1. The review is useful and informative, but surprising scant on references. There are whole antibodies and whole biomarkers with 1-2 references when there has been a lot of work performed on the antibodies
--> First of all, we appreciate the reviewer’s positive comments. As suggested, throughout this revised manuscript, we faithfully added lots of references for properly providing evidences. All added references are described in red.
2. It would be important to consider the biomarkers relation to each other, for example, what would Her2-cMet and Ron (which is not really covered) have in overall resistance by each individual biomarker.
--> We agree with the importance of biomarker relation to each other. However, different from other cancers, due to the insufficient information showing the interrelation of biomarkers in CRCs, it is hard to comprehensively cover relation among all biomarkers described in this manuscript. Further, this manuscript is focused on discussing the roles and mechanisms of monoclonal antibodies in clinical development and approved by US FDA and their cognate therapeutic targets in CRCs. Instead, as suggested by the reviewer, we added one paragraph explaining for the interrelationship of EGFR and other biomarkers such as HER2 and c-MET at the end of EGFR section (2.1; p3, line 131-142).
--> In case of recepteur d’origine nantais (RON) receptor tyrosine kinase, although several monoclonal antibodies including IMC-41A10, 29B06/07F01, and narnatumab (also known as IMC-RON8) has shown significant anti-tumor effects on pre-clinical studies, it has been known that the outcome diminish pharmaceutical enthusiasm for further development. In detail, the clinical phase I trials of narnatumab in patients with advanced solid tumors show limited activity within the dosing regimen, leading to the discontinuation of the clinical trials. Currently, antibodies targeting RON are also being used as an antibody platform for generating antibody drug conjugates. Thus, we excluded RON in this manuscript.
3. Also what about triple negative cancers? For a start, both Trastuzumab and Pertuzumab are used together with synergy https://doi.org/10.1186/bcr2888 that was later found to be due to additional tagging https://doi.org/10.1038/npjbcancer.2015.12 that deserves discussion given the partial discussion at 10.3390/cancers10100342
--> As described in the Abstract section, this manuscript has focused on extensively covering the physiological and pathophysiological roles and detailed signaling mechanisms of currently known and emerging therapeutic targets, including growth factors and their receptors as well as immune checkpoint proteins, for monoclonal antibody therapy in CRC. Additionally, we are also discussing the current status of monoclonal antibodies in clinical development and approved by US FDA for CRC therapy.
--> We guess that triple negative cancers mean triple negative breast cancers (estrogen receptor‐negative, progesterone receptor‐negative, and HER2‐negative). All references the reviewer suggested are related to breast cancers. Instead, we added one paragraph explaining for the interrelationship of EGFR and other biomarkers such as HER2 and c-MET at the end of EGFR section (2.1; p3, line 131-142).
4. There is a slight part in the beginning that discusses some antibody methods, but lacks discussion. For example can discuss further on antibody fragments e.g. https://doi.org/ 10.3389/fimmu.2017.01287 and humanization https://doi.org/10.1093/abt/ tbaa005, https:// doi.org/10.1038/s41598-017-18892-9
--> As suggested, we added the detailed description of antibody fragments and humanization methods with corresponding references in Introduction section (p1, line 37-43).
5. It would be good to consider the biomakers and therapeutics in some chronological order and how they can interact and work synergistically to aid in anticancer targeting.
--> In the revised manuscript, we changed the order of described the biomarkers and therapeutics in chronological order. For examples, for EGF/EGFR à cetuximab/panitumumab, for VEGF/VEGFRà bevacizumab/ramucirumab, for HGF/c-METà rilotumumab/ onartuzumab, for HER2à trastuzumab/pertuzumab, for CTLA-4à ipilimumab/tremelimumab, for PD-1/PD-L1à pembrolizumab/nivolumab/camrelizumab/ atezolizumab/avelumab/durvalumab. The titles of all changed section are in red. As described in Answer 2 and 3, we added one paragraph explaining for the interrelationship of EGFR and other biomarkers such as HER2 and c-MET at the end of EGFR section (2.1; p3, line 131-142).
Reviewer 2 Report
This is a welcome review of the state of the art on therapeutic monoclonal antibodies in colorectal cancer. It is well written, well organized and clear. The authors could mention work on other formats of therapeutic antibodies, such as bispecifics and ADCs, and the most advanced stage or promising formats of this class. This would render the review more complete, also in consideration that targets for bispecific sand ADCs are mostly the same as that of unconjugated monoclonals.
Author Response
Response to reviewers
We appreciate the detailed and helpful comments from the reviewers. Below, we have provided point-by-point responses to thoroughly address the concerns of each reviewer. We also performed intensive proofreading by professional proofreading company. In the revised manuscript, all of modified words and sentences are in red.
▪ Comments from Reviewer #2
1. This is a welcome review of the state of the art on therapeutic monoclonal antibodies in colorectal cancer. It is well written, well organized and clear.
--> We appreciate your positive comments.
2. The authors could mention work on other formats of therapeutic antibodies, such as bispecifics and ADCs, and the most advanced stage or promising formats of this class. This would render the review more complete, also in consideration that targets for bispecific sand ADCs are mostly the same as that of unconjugated monoclonals.
--> This review article have focused on extensively covering the known and emerging therapeutic targets and their therapeutics for monoclonal antibody therapy in CRCs. Especially, in case of therapeutics, we included monoclonal antibodies in clinical development (over phase II) and approved by FDA for CRC therapy.
--> Basically, we agree with the importance of other antibody formats including bispecific antibodies and antibody drug conjugates. However, due to the limited time for this revision, it is hard to include targets and therapeutics for bispecific antibodies and ADCs in CRCs at the same time. Instead, I added the importance of these therapeutics and additional notion in Perspective section (p16, line 631-635).
--> Further, to exclude any confusion, we corrected the title from “Recent advances in monoclonal antibody-based therapy for colorectal cancers” to “Recent advances in monoclonal antibody therapy for colorectal cancers”.
Round 2
Reviewer 1 Report
The review has significantly improved and is well good to go.
Although seen as a possible slight de-tour by the authors, I do recommend a small paragraph on discussing synergistic use of some of these mAb treatment options to improve outcomes and add value to readers that what is promising and successful in other cancers can also be adopted for treatment by colorectal cancers as the underlying mechanism is often the same. But this is optional as a value add.